# Effects of Oregano Essential Oil on IgA^+^, IgG^+^, and IgM^+^ Cells in the Jejunum of Castrated Holstein Bulls

**DOI:** 10.3390/ani13243766

**Published:** 2023-12-06

**Authors:** Qiyan Liu, Wangdong Zhang, Baoshan Wang, Jinping Shi, Pengjia He, Li Jia, Yongliang Huang, Meiling Xu, Yue Ma, Qiang Cheng, Zhaomin Lei

**Affiliations:** 1College of Animal Science and Technology, Gansu Agricultural University, Lanzhou 730070, China; liuqy@st.gsau.edu.cn (Q.L.); shijinpingyunfei@163.com (J.S.); hepengjia23@163.com (P.H.); jlyf2294@163.com (L.J.); xuml@st.gsau.edu.cn (M.X.); mayue4315@163.com (Y.M.); 2College of Animal Medicine, Gansu Agricultural University, Lanzhou 730070, China; zhangwd@gsau.edu.cn (W.Z.); wangbs2026@163.com (B.W.); 3Jing Chuan Xu Kang Food Co., Ltd., Pingliang 745000, China; chengqiang@188.com

**Keywords:** oregano essential oil, jejunum, IgA^+^ cells, IgG^+^ cells, IgM^+^ cells

## Abstract

**Simple Summary:**

Oregano essential oil is a volatile essential oil that plays an important regulatory role in various species, but whether it has an immune-boosting effect on castrated Holstein bulls has not been reported to date. Thus, the effects of oregano essential oil on IgA^+^, IgG^+^, and IgM^+^ cells in the jejuna of castrated Holstein bulls were studied. The results show that the addition of the oil did not change the diffuse distribution of IgA^+^, IgG^+^, and IgM^+^ cells, but repaired damaged intestinal epithelium, inhibited inflammation, and reduced the content of IgA+, IgG+ and IgM+ cells in the jejunum. This provides strong evidence for further research on the mechanism by which oregano essential oil improves the production performance of castrated Holstein bulls.

**Abstract:**

The aim of this study was to investigate the effect of oregano essential oil on IgA^+^, IgG^+^, and IgM^+^ cells in the jejunum of castrated Holstein bulls. Twelve castrated Holstein bulls were randomly divided into control (YCK) and oregano essential oil (YEO) groups. Pathological changes in the jejunum were observed by HE staining, and the expression levels of IgA, IgG, and IgM in the jejunum were detected by ELISA. The distributions of IgA^+^, IgG^+^, and IgM^+^ cells in the jejunum were analysed by multiplex immunofluorescence and immunohistochemistry. The results showed that the jejunal villi were detached in the YCK group, which may have been related to inflammation, while the intestinal epithelium was clear and intact in the YEO group. The expressions of IgA, IgG, and IgM were significantly reduced by 40.75%, 30.76%, and 50.87%. The IgA^+^, IgG^+^, and IgM^+^ cells were diffusely distributed in the lamina propria of the jejunum, and were reduced by 17.07%, 6.44%, and 6.15%, respectively. Oregano essential oil did not alter the distribution characteristics of IgA^+^, IgG^+^, or IgM^+^ cells in the jejunum, but it suppressed inflammatory response, decreased immunoglobulin content, and significantly enhanced the formation of an immune barrier in the gastrointestinal mucosa.

## 1. Introduction

In humans and animals, the small intestine is composed of three segments: the duodenum, jejunum, and ileum. Jejunal peristalsis is fast, and the intestine is often emptied, which is an important part of the digestion and absorption of nutrients. The jejunum is an important part of the intestinal mucosal immune system: the intestinal epithelium and mucus layer constitute the physical barrier of the immune system of the intestinal mucosa. If this barrier is disrupted, pathogens invade the gut, and the immune system develops a protective immune response, such as inducing immunoglobulin production against the pathogen [1]. Immunoglobulins play key roles in protecting the intestine from inflammation by regulating the intestinal flora composition [2], inhibiting the inflammatory response, blocking the adsorption of pathogenic bacteria onto the mucosa, and regulating and maintaining the intestinal immune system [3]. Immunoglobulins also have a blocking effect on certain antigenic substances ingested through food, such that these antigens are free from the mucosal surface and do not enter the body to exert an immune defence.

Immunoglobulin is an antibody-active animal protein present in the plasma or tissues that is produced in the body when stimulated by antigens [4]. It can specifically bind antigens to form antigen–antibody complexes, thereby eliminating the harmful effects of pathogenic bacteria on the body. IgA mainly exists in the form of dimers [5], which can bind to the corresponding antigen to activate complement-solubilising antigens. It is an important protein for the body’s anti-infection immunity [6], forming the first protective barrier for the mucosal surface. IgG is the most abundant immunoglobulin with the highest content in the body, which can effectively resist bacteria [7] and viruses [8]. It can improve the phagocytosis of cells [9], thereby blocking the corresponding antigens from entering tissues through the mucosa and forming a protective barrier on the surface of the mucosa. IgM is the largest molecular weight immunoglobulin and exists mainly in pentamer form [10]. It can exert immune effects, such as bacteriolytic effects and neutralising viruses by activating complement [11], and its agglutination and haemolysis with complement participation are 500–1000 times higher than those of IgG, making it a highly potent antibody. Therefore, IgA, IgG, and IgM play important roles in jejunal immunity.

Oregano essential oil (OEO) is a volatile essential oil extracted from oregano grass. Its main components are carvacrol and thymol [12]. OEO can play an important regulatory role among different species, improve the reproductive performance of sows, and play a significant role in piglet weaning, intestinal microbiota improvement, and nutrient absorption [13,14,15]. It has no effect on rumen pH in sheep but can inhibit the growth of protozoa [16] by improving the antioxidant status of broilers, alleviating local oxidative stress in the intestine, regulating the composition of microflora, and improving digestibility [17,18,19]. The addition of OEO to the diet can also improve rumen digestion and the antioxidant capacity of Pingliang Red Cattle [20,21], thereby promoting nutrition and improving the meat quality of beef and improving the meat and fibre quality of Hexi cashmere goats [22]. However, there are no reports on whether OEO has an immune-boosting effect in castrated Holstein bulls. Therefore, this study explored the content of immunoglobulins and the number of corresponding plasma cells in castrated Holstein bulls after the addition of OEO. It provides strong evidence for further research on the mechanism by which OEO improves the production performance of castrated Holstein bulls.

## 2. Materials and Methods

### 2.1. Test Materials

The basal feed and test animals were provided by the Huarui Company in Minle County, Zhangye, Gansu Province, China. The oregano essential oil used for the test was purchased from Ralco Inc. (Marshall, MN, USA) and contained 1.3% oregano essential oil and 98.7% natural food-grade inert carrier.

### 2.2. Experimental Design and Feeding Management

A total of 18 castrated Holstein bulls with good health and a weight of (350.30 ± 6.49) kg were selected around 10 months of age. They were randomly divided into 2 groups, namely control group (YCK group) and oregano essential oil group (YEO group), with 3 replicates in each group and 3 cattle in each replicate. The YCK group was fed a basal diet, and the YEO group was fed with 20 g/(head·d) of oregano essential oil on the basal diet. The test period was 300 days, the adaptive feeding was 30 days, and the main test period was 270 days: a total of 9 months (I~IX). In accordance with the NRC (2016) beef cattle nutritional requirements for target daily weight gain, nutritional standards of the lowest cost and optimal efficiency, and the design of full-price complete feeds by stage, each stage of the basic ration composition and nutrient levels are shown in Table 1. Prior to the start of the experiment, the animal enclosures were thoroughly disinfected, test numbers were worn, and the cows were fed once a day at 07:00 and once a day at 17:00. Twenty grams of oregano essential oil was weighed, mixed thoroughly with concentrate, and spread on the feed surface to ensure complete intake of feed by the cows, which were fed and watered ad libitum during this period. At the end of the feeding trial, 6 cows from each group were randomly selected for slaughter, and 24 h of food and 2 h of water were prohibited before slaughter. A jejunal tissue sample was collected, and the mucosal surface was thoroughly washed with normal saline and fixed with 4% neutral paraformaldehyde solution for hematoxylin–eosin (HE) staining, immunohistochemistry, and multiplex immunofluorescence assays. A second jejunal molecular sample was collected, placed in a sterile cryopreservation tube, and transferred to a −80 °C freezer for storage for enzyme-linked immunosorbent assay testing.

### 2.3. Hematoxylin–Eosin (HE) Staining

Paraffin sections (4 μm) were routinely prepared and dewaxed to water. Hematoxylin staining was performed for 6 min (Item No. H8070, Soleberg technology Co., Ltd., Beijing, China), then the sections flushed with tap water for 30 min. Hydrochloric acid–alcohol differentiation was performed for 10 s, followed by flushing with tap water for 30 min to return to blue. Eosin staining was performed for 8 min (Item No. G1102, Soleberg technology Co., Ltd., Beijing, China), and gradient alcohol dehydration was then performed with neutral gum seal. See references for detailed steps [23]. Finally, the jejunum was photographed using a pathology slide scanner (Model No. DX1, Sruidi Medical Technology Co., Ltd., Jinan, China) in order to observe the pathological changes in the jejunum.

### 2.4. Enzyme-Linked Immunosorbent Assay

Frozen tissue samples were thawed on ice, 1.0 g of tissue was accurately weighed, and 1 mL PBS and two magnetic beads were added. This was then homogenised at −10 °C for 15 min, and centrifuged for 10 min (4 °C, 12,000 rpm). The supernatant was then collected, and the protein concentration determined using a BCA protein assay kit (Cat#PC0020, batch numbers 20, 210, 908, Solarbio technology Co., Ltd, Beijing, China). The immunoglobulin expression in the jejunum was determined using an enzyme-linked immunoassay kit, and ELISA was used to detect the immunoglobulin expression levels in each segment of the jejunum (bovine IgA (Item no. YJ542063), bovine IgG (Item no. YJ330698), and bovine IgM (Item no. YJ152690) Enzyme-Linked Biotechnology Co, Ltd., Shanghai, China). The inter- and within-assay coefficients of variation for the ELISA were <10% and 15%, respectively, and the test results were considered reliable.

### 2.5. Multiplex Immunofluorescence

A four-colour multiple fluorescence staining kit (Item no. RC0086-34R, Rutron Biotechnology Co., Ltd., Shanghai, China) was used to fluorescently label IgA^+^, IgG^+^, and IgM^+^ cells in the same jejunal paraffin section using tyramide signal amplification. The sections were de-paraffinized in water, incubated with primary anti-IgM antibodies (Item no. ab112752, abcam, Waltham, MA, USA) at a working concentration of 1:200, and TY570 fluorescent dye was added for 2–15 min. A second round of incubation was performed using IgG antibodies (Item no. ab69272, abcam, Waltham, MA, USA) at a working concentration of 1:600, TY690 fluorescent dye was added for 2–15 min. Finally, a third round of incubation was performed with dros of IgA antibodies (Item no. ab1126302, abcam, Waltham, MA, USA) at a working concentration of 1:200 and TY520 fluorescent dye was added for 2–15 min. The non-covalently bound antibody was washed away using the thermal repair method and sealed with a BSA sealing solution for 30 min. Subsequently, the nuclei of the cells were re-stained with DAPI (Item No. C0065, Soleberg technology Co., Ltd., Beijing, China) and sealed with an anti-fluorescence quencher for observation. Immunofluorescent sections were photographed using luorescence microscope (Model No. VS200, Olympus Corporation, Tokyo, Japan) f to obtain high-quality fluorescence images of all fluorescence channels and the distribution patterns of IgA^+^, IgG^+^, and IgM^+^ cells were observed.

### 2.6. Immunohistochemical Staining

Paraffin sections (4 μm) were routinely prepared and de-paraffinized to water. The sections were treated with 3% H_2_O_2_ for 10 min to eliminate endogenous peroxidase. The antigen was repaired using the thermal repair method (0.8 g citric acid with 6 g trisodium citrate), and 5% BSA was added to close the sections for 30 min at 37 °C. The optimal working concentrations for incubation of the primary antibody IgA (Item no. ab112630, abcam, Waltham, MA, USA), IgG (Item no. ab6927, abcam, Waltham, MA, USA), and IgM (item no. ab112752, abcam, Waltham, MA, USA) were 1:200, 1:600, and 1:200, respectively. A secondary anti-SABC mouse/rabbit IgG kit (Item no. SA1020) was used, which is a ready-to-use kit. Detailed steps can be found in [24]. Immunohistochemical sections were scanned and photographed with a pathology section scanner (Model No. DX1, Sruidi Medical Technology Co., Ltd., Jinan, China). Ten fields of view were randomly intercepted from each section, IgA^+^, IgG^+^, and IgM^+^ cells in the fields of view were counted, and their respective densities were calculated using Image-ProPlus 6.0.

### 2.7. Statistical Analysis

The *t*-test of independent samples was used to analyse the expression and quantity changes of immunoglobulins between the YCK and YEO groups. Duncan’s multiple test was used to analyse the quantity changes of three kinds of immunoglobulins in each group, and the results of the experiments were expressed as the mean ± SD (*p* < 0.05: significant difference; *p* > 0.05: insignificant difference).

## 3. Results

### 3.1. Pathologic Jejunum Changes in the YCK and YEO Groups

In this study, it was found by HE staining that the jejunal intestinal villi in the YCK group were severely damaged and almost completely detached, the local central celiac duct was filled with a pink plasma substance, and the central celiac duct was dilated; more neutrophil infiltration was seen around the intestinal line, and there was obvious edema within the lamina propria of the intestinal villi (Figure 1A–F). In contrast, in the YEO group, the intestinal villi were clear and intact in structure, the localized central celiac duct was not dilated, no neutrophil infiltration was seen, and there was no edema within the lamina propria (Figure 2A–F).

### 3.2. Expression of IgA, IgG, and IgM in the Jejunum

ELISA results showed that the expression of IgA, IgG, and IgM in the jejunum gradually decreased after the addition of OEO, with IgA, IgG, and IgM decreasing by 40.75, 30.76, and 50.87%, respectively (Table 2). The expression of IgA, IgG, and IgM in the YEO group was significantly lower than that in the YCK group (*p* < 0.05) (Figure 3).

### 3.3. Distribution Pattern of IgA^+^, IgG^+^, and IgM^+^ Cells in the Jejunum

In the jejunum, IgA^+^, IgG^+^, and IgM^+^ cells were distributed diffusely in the intestinal lamina propria (Figure 4), and there was no significant difference in the distribution of positive cells between the YEO and YCK groups.

### 3.4. Number Changes of IgA^+^, IgG^+^, and IgM^+^ Cells in the Jejunum

According to the quantitative statistics of the immunohistochemical tests (Table 3), the distribution densities of IgA^+^ cells in the jejuna of the YEO and YCK groups were 66.49 and 80.18 cells/10^4^ μm^2^, respectively; the distribution densities of IgG^+^ cells in the jejuna of the YEO and YCK groups were 72.80 and 77.80 cells/10^4^ μm^2^, respectively; and the distribution densities of IgM^+^ cells in the jejuna of the YEO and YCK groups were 59.63 and 63.54 cells/10^4^ μm^2^, respectively. The numbers of IgA^+^, IgG^+^, and IgM^+^ cells in the YEO group were significantly lower than those of the YCK group (Figure 5, Figure 6 and Figure 7). The IgA^+^, IgG^+^, and IgM^+^ cell densities were significantly (*p* < 0.05) decreased by the addition of OEO to the diet, with 17.07%, 6.44%, and 6.15% decreases, respectively (Figure 8A–C). The YCK group had the highest total density of IgA^+^, IgG^+^, and IgM^+^ cells (Figure 8D). 

### 3.5. Differences between IgA^+^, IgG^+^, and IgM^+^ Cells

The distribution densities of IgA^+^, IgG^+^, and IgM^+^ cells at the same site in the YCK group were highest for IgA^+^ cells, followed by IgG^+^ and then IgM^+^ cells; the differences were significant (*p* < 0.05) between IgA^+^ and IgM^+^ cells, as well as between IgG ^+^ and IgM^+^ cells, whereas the differences between the IgA^+^ and IgG ^+^ cells were not significant (*p* > 0.05) (Figure 9A).

The distribution densities of IgA^+^, IgG^+^, and IgM^+^ cells at the same site in the YEO group were highest for IgG^+^ cells, followed by IgA^+^ and then IgM^+^ cells. Significant (*p* < 0.05) differences were observed among IgA^+^, IgG^+^, and IgM^+^ cells in the YEO group (Figure 9B).

## 4. Discussion

Owing to its direct communication with the external environment, the intestine is constantly exposed to many potential antigens while absorbing nutrients. This study showed that depopulated Holstein bulls in the YCK group suffered from inflammation, resulting in significant localised damage to the jejunal mucosa. The addition of excessive concentrate to the diet is likely to induce gastrointestinal damage, causing inflammatory response [25,26]. Stress is another factor that induces intestinal damage and causes inflammation, such as heat stress, which leads to increased intestinal permeability, localised inflammation, and the loss of many intestinal epithelial cells, leading to intestinal villi damage [27,28,29]. Cold stress causes oxidative stress and inflammatory injury, leading to a significant decrease in the height of intestinal villi and so on [30,31]. In addition, recent studies have reported that intestinal flora disruption is a key factor in inducing intestinal injury. The microbial flora in the jejunum mainly includes the thick-walled phylum, Proteobacteria, the Actinobacteria phylum, and the Mycobacterium phylum; these pathogenic bacteria can damage the immune system, leading to damage to the intestinal barrier, and can also trigger inflammation [32,33]. Blumberg et al. [34] demonstrated that a decrease in the thick-walled phylum was one of the most important manifestations of the occurrence of intestinal inflammation; the Proteobacteria belongs in the category of pathogenic bacteria, and several studies have proven that intestinal inflammation is related to an increase in the Proteobacteria [35,36]. Therefore, jejunal mucosal damage in the YCK group may be related to the disturbance of the colonising flora of the digestive tract induced by the above stimuli.

The GI mucosal immune system is central to the regulation of immune homeostasis [37,38]. Immune molecules such as IgA, IgG, and IgM are important defence molecules in the GI mucosal immune barrier. In this study, the expression levels of IgA, IgG, and IgM in the jejunum were sequentially reduced by the addition of OEO, and the difference was significant (*p* < 0.05); the numbers of IgA^+^, IgG^+^, and IgM^+^ cells distributed within the lamina propria of the jejunum were also synchronously reduced, and the difference was significant (*p* < 0.05). It is suggested that oregano essential oil can significantly reduce the number of IgA^+^, IgG^+^, and IgM^+^ cells and protein expression levels in the jejunum of depopulated Holstein bulls. Some studies show that the expression levels of IgA, IgG, and IgM are elevated after the organism is infected with a virus by inducing a certain immune response [39,40,41]; the main components of oregano essential oil are carvacrol and thymol, which belong to volatile fatty acids in terms of their chemical structure. Current studies confirm that volatile fatty acids will effectively slow down the inflammatory response [42,43], promote intestinal mucosal proliferation and maintain intestinal mucosal morphology, and inhibit lymphocyte activation and proliferation and myeloperoxidase activity in neutrophils. Venkatraman et al. [44] found that butyrate increases colonic mucosal permeability and reduces neutrophil infiltration in an animal model of colitis. Therefore, in conjunction with the HE histopathology results, we hypothesised that the cause of the decreased immunoglobulin content in the YEO group may be related to the ability of these two components of oregano essential oil to inhibit the inflammatory response.

The distribution pattern of the three immunoglobulins in the jejunum of the YCK group contained the highest density of IgA^+^ cells, followed by IgG^+^ cells, and IgM^+^ cells, while the YEO group had the highest density of IgG^+^ cells, followed by IgA^+^ cells and IgM^+^ cells. It is suggested that the addition of oregano essential oil to the diet had the most drastic effect on IgA^+^ cells, followed by IgG^+^ and IgM^+^ cells. The formation of different types of antibody-secreting cells in the digestive tract is mainly related to the combined regulation of multiple factors, such as intestinal flora, Tfh cell auxiliary in the intestinal lymphoid collector [45], antigenic class-transformation regulation [46,47], and cytokine (such as IL-8, IL-4, and TGF-β)-induced [48,49,50]. Further investigation is needed to determine the detailed mechanisms of the effect of oregano essential oil on IgA^+^, IgG^+^, and IgM^+^ cytogenesis regulation.

## 5. Conclusions

The results of this study showed that the three positive cells were diffusely distributed in the intestinal lamina propria. The addition of OEO to the diets of castrated Holstein bulls did not alter this distribution, but rather enhanced the immunity of these bulls, repaired the damaged immune system, reduced the inflammatory response, and reduced immunoglobulin content. This study provides a theoretical basis for OEO to regulate immunoglobulin content and suppress inflammatory responses. 

## Figures and Tables

**Figure 1 animals-13-03766-f001:**
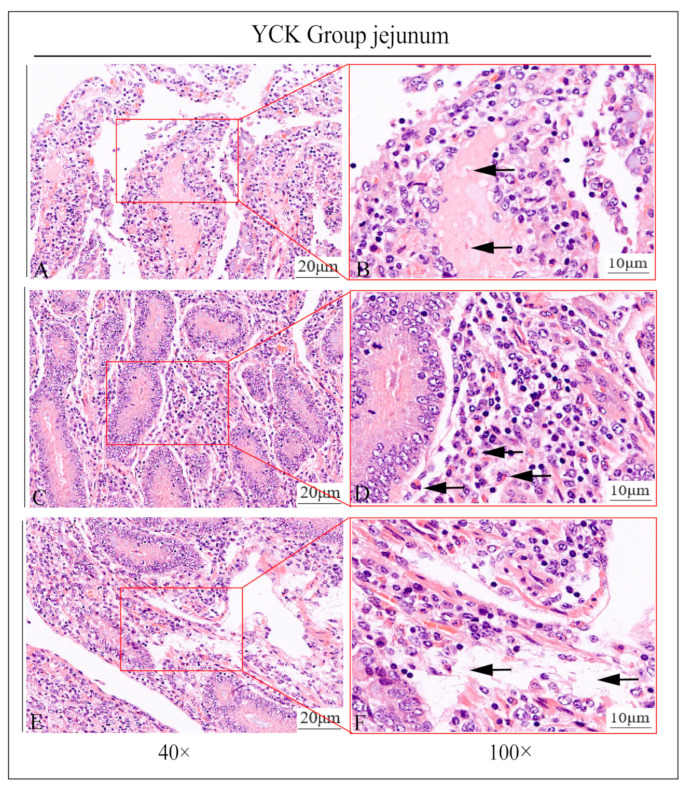
Partial view of jejunum of castrated Holstein bulls in YCK group. (**A**,**B**) arrows in the figures show dilatation of the central celiac duct; (**C**,**D**) arrows in the figures show neutrophil in-filtration; (**E**,**F**) arrows in the figures show edema of the lamina propria. The right figure (scale bar 10 μm) is a partial enlargement of the left figure (scale bar 20 μm).

**Figure 2 animals-13-03766-f002:**
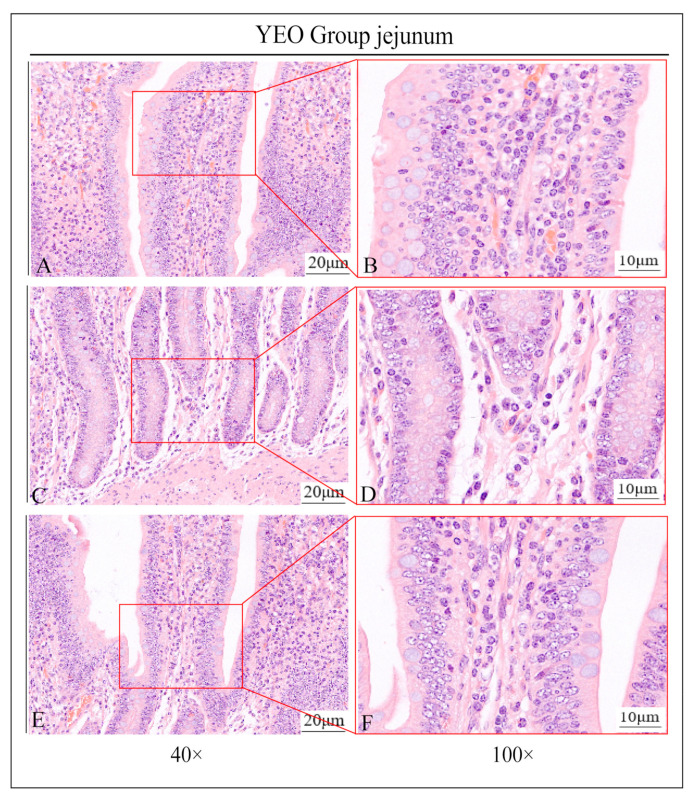
Partial view of jejunum of castrated Holstein bulls of YEO group. (**A**,**B**) show no dilatation of the central celiac duct; (**C**,**D**) show no infiltration of neutrophils; (**E**,**F**) show no edema of the lamina propria. The right panel (scale bar 10 μm) is a partial enlargement of the left panel (scale bar 20 μm).

**Figure 3 animals-13-03766-f003:**
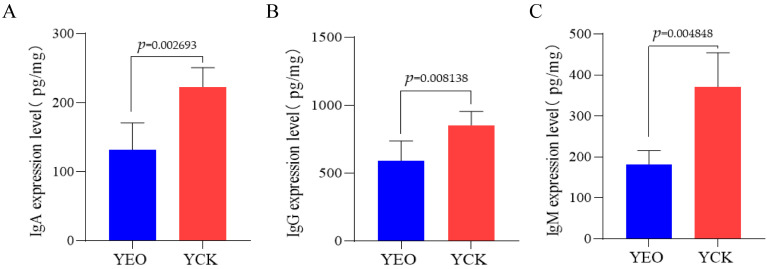
Effect of adding oregano essential oil on the expression levels of IgA, IgG, and IgM in the jejunum of depopulated Holstein bulls. (**A**) Effect of adding oregano essential oil on IgA expression level. (**B**) Effect of adding oregano essential oil on IgG expression level. (**C**) Effect of adding oregano essential oil on IgM expression level. *p* < 0.05 indicates significant difference.

**Figure 4 animals-13-03766-f004:**
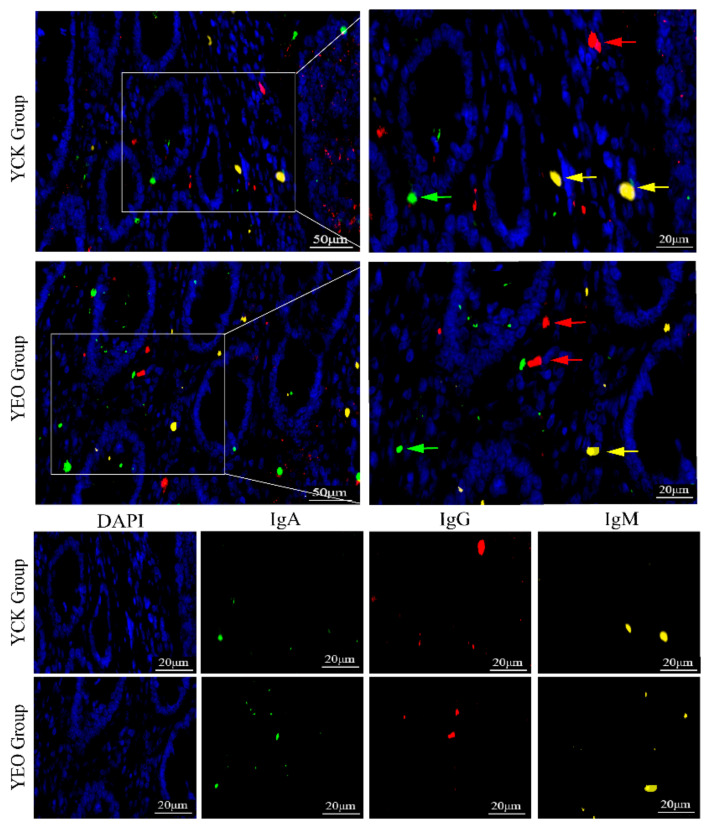
Graph of the results of IgA/IgG/IgM/DAPI trichrome fluorescence staining in jejunum. DAPI stained the nuclei of the cells in blue color, and the green arrows indicated the IgA^+^ cells, the red arrows indicated the IgG^+^ cells, and the yellow arrows indicated the IgM^+^ cells. The right image (scale bar, 20 μm) is a partial enlargement of the left image (scale bar, 50 μm).

**Figure 5 animals-13-03766-f005:**
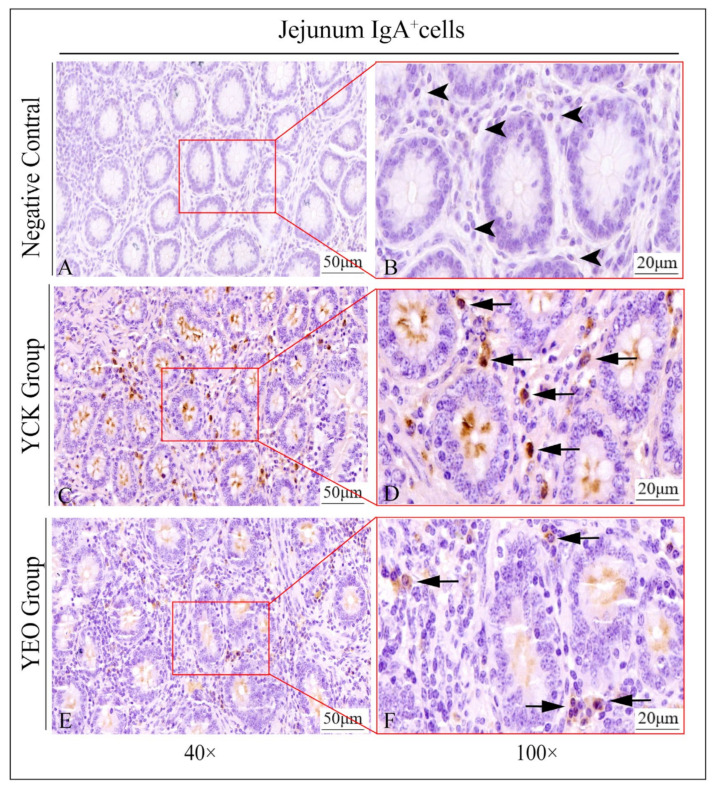
Distribution of IgA^+^ cells in the jejunum of depopulated Holstein bulls. (**A**,**B**) are negative controls; (**C**,**D**) are controls; (**E**,**F**) are oregano essential oil groups. Triangular arrows indicate plasma cells, arrowheads indicate IgA^+^ cells, and the right image (scale bar, 20 μm) is a local magnification from the left image (scale bar, 50 μm).

**Figure 6 animals-13-03766-f006:**
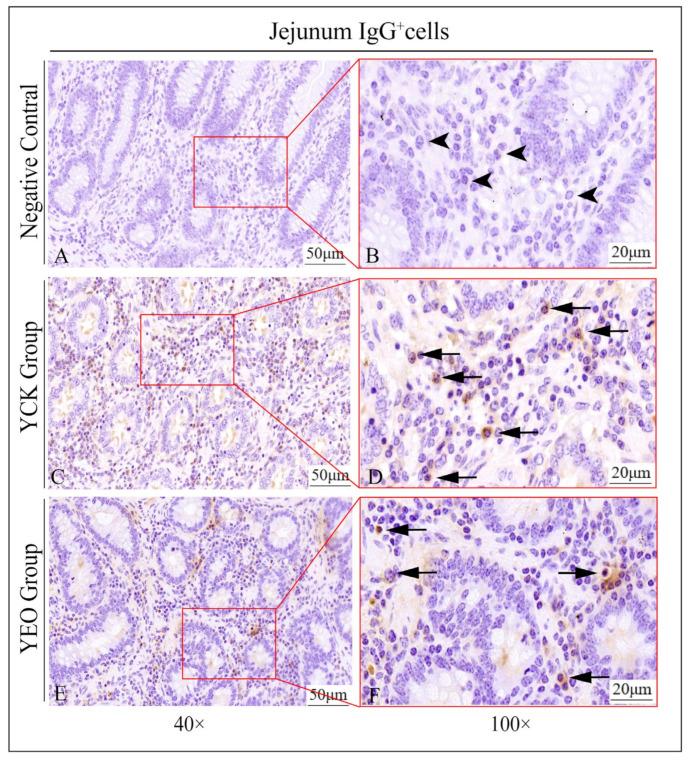
Distribution of IgG^+^ cells in the jejunum of depopulated Holstein bulls. (**A**,**B**) are negative controls; (**C**,**D**) are controls; (**E**,**F**) are oregano essential oil groups. Triangular arrows indicate plasma cells, arrowheads indicate IgG^+^ cells, and the right image (scale bar, 20 μm) is a local magnification from the left image (scale bar, 50 μm).

**Figure 7 animals-13-03766-f007:**
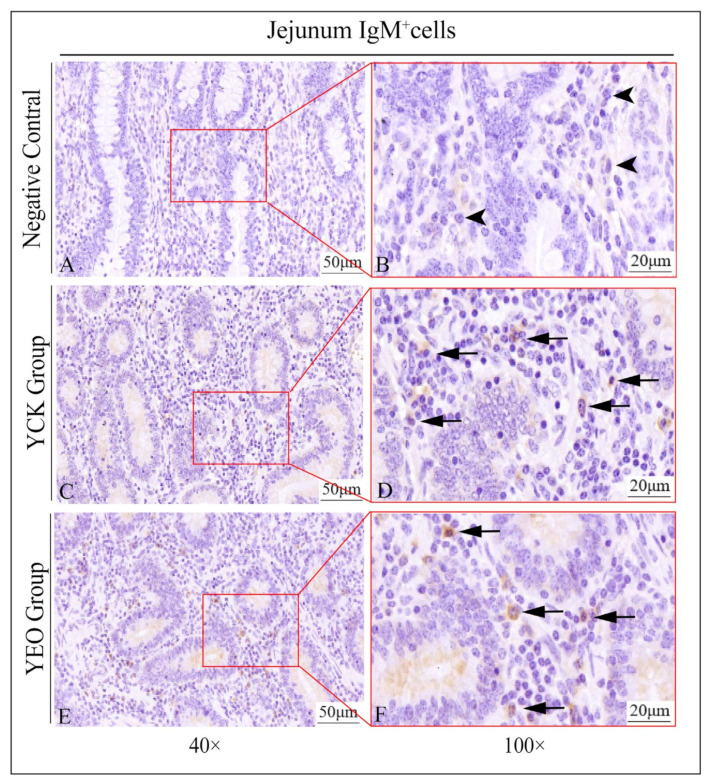
Distribution of IgM^+^ cells in the jejunum of depopulated Holstein bulls. (**A**,**B**) are negative controls; (**C**,**D**) are controls; (**E**,**F**) are oregano essential oil groups. Triangular arrows indicate plasma cells, arrowheads indicate IgM^+^ cells, and the right image (scale bar, 20 μm) is a local magnification from the left image (scale bar, 50 μm).

**Figure 8 animals-13-03766-f008:**
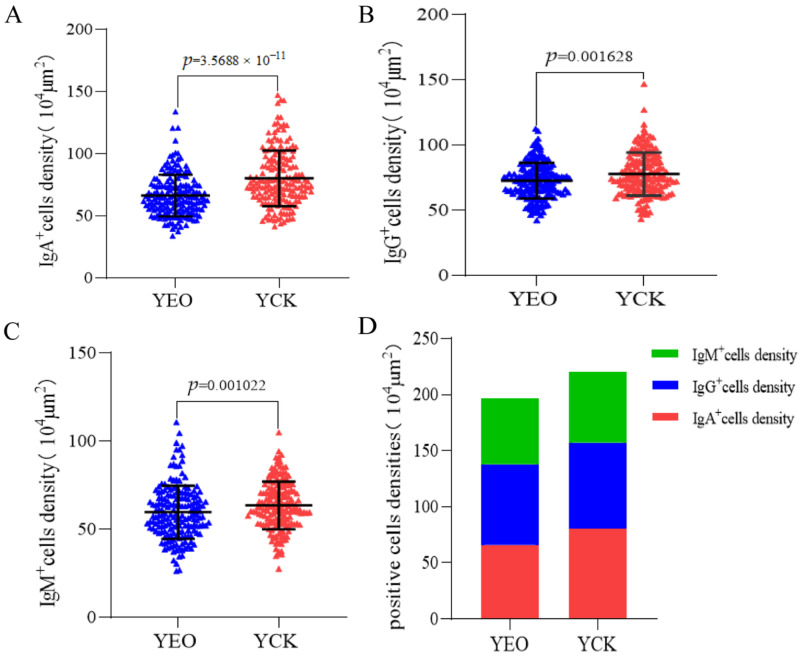
Effect of addition of oregano essential oil on IgA^+^, IgG^+^, and IgM^+^ cell densities in the jejunum. (**A**) Effect on IgA^+^ cell density after addition of oregano essential oil. (**B**) Effect on IgG^+^ cell density after addition of oregano essential oil. (**C**) Effect on IgM^+^ cell density after addition of oregano essential oil. (**D**) Total density of positive cells in the oregano essential oil group versus the control group. *p* < 0.05 indicates significant difference.

**Figure 9 animals-13-03766-f009:**
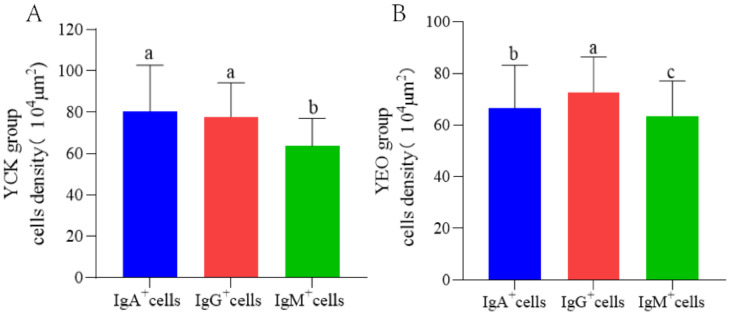
Differences between IgA^+^, IgG^+^, and IgM^+^ cells. (**A**) Differences between IgA^+^, IgG^+^, and IgM^+^ cells in the YCK group; (**B**) Differences between IgA^+^, IgG^+^, and IgM^+^ cells in the YEO group, with different letters indicating significant differences *p* < 0.05; with the same letters indicating non-significant differences *p* > 0.05.

**Table 1 animals-13-03766-t001:** Composition and nutrient levels of the basal feed.

Items%	Months				
	I	II	III	IV	V~IX
Ingredients					
Corn silage	45.00	40.00	30.00	25.00	20.00
Whole cottonseed	0.00	0.00	0.00	0.00	10.78
Flattened corn	14.84	10.00	8.00	5.00	10.00
Corn	25.17	41.36	51.06	59.93	54.36
Soybean meal	8.21	3.20	0.00	2.00	0.00
Rapeseed meal	0.00	0.54	5.17	3.57	0.00
Cottonseed meal	0.00	2.00	2.00	1.00	1.00
Pea protein powder	3.49	0.00	0.00	0.00	0.00
Fatty acid calcium	0.00	0.00	0.00	0.00	1.00
CaHPO_4_	1.12	0.80	1.10	1.28	0.65
Na Cl	0.62	0.30	0.47	0.50	0.47
NaHCO_3_	1.11	1.00	1.28	1.00	1.15
MgO	0.00	0.19	0.28	0.16	0.13
Premix ^1^	0.44	0.61	0.64	0.56	0.46
Total	100.00	100.00	100.00	100.00	100.00
Nutrition levels ^2^					
Crude Protein	11.94	10.95	10.80	10.60	9.80
Total digestible nutrients	72.10	76.54	77.51	78.74	80.52
Net energy maintenance/(MJ/kg)	7.44	7.97	8.14	8.31	8.53
Net weight gain/(MJ/kg)	4.92	5.31	5.47	5.61	5.80
Ca	0.61	0.50	0.59	0.63	0.37
P	0.31	0.33	0.36	0.35	0.34

Note: ^1^ Contained the following per kg of premix: VA 4,000,000 IU, VD 3,600,000 IU, VE 120,000 IU, Mn 3340 mg, Fe 4800 mg, Zn 12,600 mg, Cu 3140 mg, Se 60 mg, I 90 mg, Co 22 mg, nicotinic acid 5970 mg, biotin 200 mg. ^2^ Total digestible nutrients, net energy maintenance, and net weight gain were calculated values, while the others were measured values.

**Table 2 animals-13-03766-t002:** Expression levels of IgA, IgG, and IgM in the jejunum.

Items	YEO	YCK	*p* Value	Decrease Rate
IgA(pg/mg)	132.12 ± 38.64	222.98 ± 27.61	0.002693	40.75%
IgG(pg/mg)	591.95 ± 146.78	854.90 ± 101.13	0.008138	30.76%
IgM(pg/mg)	182.30 ± 33.48	371.04 ± 83.18	0.004848	50.87%

Note: This result was obtained from an independent samples *t*-test, *p* < 0.05, indicating a significant difference. Decrease rate = (X − Y)/X*100%, X is positive cells in control group; Y is positive cells in oregano essential oil group.

**Table 3 animals-13-03766-t003:** Changes in the amount of IgA^+^, IgG^+^, and IgM^+^ in the jejunum.

Items	YEO	YCK	*p* Value	Decrease Rate
IgA^+^ cells/10^4^ μm^2^	66.49 ± 16.67	80.18 ± 22.48	3.5688 × 10^−11^	17.07%
IgG^+^ cells/10^4^ μm^2^	72.80 ± 13.68	77.80 ± 16.47	0.001628	6.44%
IgM^+^ cells/10^4^ μm^2^	59.63 ± 15.06	63.54 ± 13.57	0.001022	6.15%

Note: This result was obtained from an independent samples *t*-test, *p* < 0.05, indicating a significant difference. Decrease rate = (X − Y)/X*100%, X is positive cells in control group; Y is positive cells in oregano essential oil group.

## Data Availability

The data presented in this study are available on request from the corresponding author.

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
