# Peer review of "Effects of Oregano Essential Oil on IgA+, IgG+, and IgM+ Cells in the Jejunum of Castrated Holstein Bulls"

_animals, 2023, doi:10.3390/ani13243766_

Round 1
Reviewer 1 Report
Comments and Suggestions for Authors
In this manuscript, the authors have investigated the effects of oregano essential oil (OEO) on IgA+, IgG+, and IgM+ cells in the jejunum of castrated Holstein bulls. While I appreciate the effort of the work presented, I would like the authors to consider the following correction.
1. In this manuscript, whether it is persuasive to judge inflammation in castrated bulls based on elevated immune indicators? Generally, we believe that an increase in immunoglobulin can be considered a positive response, indicating that immune system is actively responding to infections or diseases.
2. Sufficient evidence about inflammation caused by castration of bulls should be added in introduction section.
3. The main components of oregano essential oil (OEO) are agarol, carvacrol, and thymol. Whether the authors analyzed the content of agarol, carvacrol, and thymol in OEO used in this experiment?
4. The authors need to provide more comprehensive discussion rather than only describing your results, especially in the second paragraph of discussion section.
5. Bovine immunoglobulins also include IgD and IgE. Why does the manuscript only focus on IgA, IgG, and IgM? Is there any evidence indicating that these three immunoglobulins are more important or have certain traits for castrated bulls?
6. In conclusion section, the authors emphasized that oregano essential oil (OEO) regressed the immunoglobulin content towards a normal level, which seem to be inaccurate, because there were no non-castrated bull cells as healthy controls.
Comments on the Quality of English Language
There are some language issues in this manuscript, I strongly recommend modifying the language with assistance of native speakers.
Author Response
Dear Reviewer:
Thank you for your review and suggestions on our manuscript entitled "Effects of oregano essential oil on IgA+, IgG+ and IgM+ cells in the jejunum of castrated Holstein bulls" (ID: animals-2725639). All these comments are valuable and helpful for us to revise and improve the paper. We have carefully studied these comments and revised them. We hope that they will be recognized. The revised parts are highlighted in the paper with yellow background.
The revised manuscript has been uploaded for your review.
Thank you and with the utmost respect.
Qiyan Liu
The major revisions in the paper and responses to your comments are listed below:
1.In this manuscript, whether it is persuasive to judge inflammation in castrated bulls based on elevated immune indicators? Generally, we believe that an increase in immunoglobulin can be considered a positive response, indicating that immune system is actively responding to infections or diseases.
Response: Thank you very much for your valuable comments. Normally, within the physiologic range, elevated immunoglobulin levels are an indication of the body's ability to fight pathogenic microorganisms. However, in the present study, the immunoglobulin content in the uncontrolled group had exceeded the normal physiologic range by about two times [1]. Secondly, in the present study, the histopathological observations revealed the presence of more obvious damage in the jejunum of depopulated Holstein cows, which was clearly modified in the corresponding position of the paper. It has been shown that with the onset of infection there is a corresponding increase in immunoglobulins, e.g., Ruytinx Pieter [2] in a study on neocoronary pneumonia showed that the more severe the disease the higher the IgG and IgM levels. Therefore, the present study concluded that there was damage to the jejunum of control cows, manifested by higher immunoglobulin levels, and that there was a more pronounced mitigating effect of oregano essential oil on this damage.
References:
1.Arevalo, M.T.; Rizzo, G.M.; Polsky, R.; Glaros, T.; Mach, P.M. Proteomic characterization of immunoglobulin content in dermal interstitial fluid. J. Proteome Res. 2019, 18, 2381-2384.
2.Pieter, R.; Patrick, V.; Judith, F.; Zoë, P.; Stef, T.; Niels, H.; Piet, S.; Ina, C.; Joris, P.; Karolien, V.; Davy, K.; Jean-Luc, R.; Tom, V.; Peter, M.; Björn, S.; Dieter, M.; Veerle, S. Comprehensive antibody and cytokine profiling in hospitalized COVID-19 patients in relation to clinical outcomes in a large Belgian cohort. SCI REP-UK 2023.
2.Sufficient evidence about inflammation caused by castration of bulls should be added in introduction section.
Response: Thank you very much for your valuable comments. Histopathology is one of the common means to observe tissue lesions. In this study, we found that the jejunal intestinal villi in the YCK group were more severely damaged by HE staining, the local central celiac duct was filled with pink plasma material central celiac duct was dilated, more neutrophil infiltration was seen around the local intestinal glands as well as some intestinal villi had obvious edema within the lamina propria. And the immunoglobulin level was too high in the control group, and these reasons allowed us to determine that there was significant damage in the intestinal tract of depopulated Holstein cows. However, there are more reasons for this change, among which overfeeding of concentrates is one of the evidences of damage [1]. It is possible that the fattening was accompanied by damage to the intestinal tract of depopulated Holstein cattle, and it remains to be investigated whether castration can cause inflammation.
References:
1.de Andrade, P.V.; Schmidely, P. Effect of duodenal infusion of trans10, cis12-cla on milk performance and milk fatty acid profile in dairy goats fed high or low concentrate diet in combination with rolled canola seed. Reprod Nutr Dev 2006, 46, 31-48.
3.The main components of oregano essential oil (OEO) are agarol, carvacrol, and thymol. Whether the authors analyzed the content of agarol, carvacrol, and thymol in OEO used in this experiment?
Response: Thank you very much for your valuable comments. the oregano essential oil used in this study was purchased from (Ralco Corporation, Marshall, Minnesota, USA) containing 1.3% oregano essential oil and 98.7% natural feed grade inert carrier. This brand of oregano essential oil has been published in our group [1,2].
References:
1.Jia, L.; Wu, J.; Lei, Y.; Kong, F.; Zhang, R.; Sun, J.; Wang, L.; Li, Z.; Shi, J.; Wang, Y.; Wei, Y.; Zhang, K.; Lei, Z. Oregano essential oils mediated intestinal microbiota and metabolites and improved growth performance and intestinal barrier function in sheep. Front Immunol 2022, 13, 908015.
2.Zhang, R.; Wu, J.; Lei, Y.; Bai, Y.; Jia, L.; Li, Z.; Liu, T.; Xu, Y.; Sun, J.; Wang, Y.; Zhang, K.; Lei, Z. Oregano essential oils promote rumen digestive ability by modulating epithelial development and microbiota composition in beef cattle. Front Nutr 2021, 8, 722557.
4.The authors need to provide more comprehensive discussion rather than only describing your results, especially in the second paragraph of discussion section.
Response: Thank you very much for your valuable comments. The discussion section has been modified in the article and highlighted with a yellow background. I've added the revised discussion section here for your viewing pleasure.
In this study, we observed the pathological changes of the jejunum by HE staining and found that the intestinal epithelium of the jejunum in the YCK group was severely damaged, resulting in the shedding of most of the intestinal villi; the localized central celiac ducts were filled with pink plasma material, and the central celiac ducts were dilated; a large number of neutrophils could be seen infiltrating around the intestinal line; and the intestinal villi lamina propria was obviously edematous, which was probably caused by inflammation. Conversely, the intestinal epithelium of the YEO group was structurally intact, suggesting that the addition of OEO to the diet inhibited inflammatory reactions and repaired the damaged intestinal epithelium. The ELISA detection revealed that the expression levels of IgA, IgG, and IgM in the jejunum decreased sequentially after the addition of OEO and that the expression of immunoglobulins in the YEO group was significantly lower than that in the YCK group. Observation of the distribution patterns of IgA+, IgG+ and IgM+ cells using multiple immunofluorescence techniques revealed that the three positive cells were diffusely distributed in the intestinal lamina propria. Diffuse distribution is the most significant morphological feature of the immune defence system of the intestinal mucosa, which helps positive cells secrete corresponding antibodies to form a complete protective barrier. It has been shown that volatile fatty acids can play a mitigating role in inflammation[1,2]. The main components of oregano essential oil are carvacrol and thymol, which also belong to the volatile fatty acids in terms of their chemical structure, and therefore we hypothesize that the cause of the decrease in the immunoglobulin content of the YEO group may be related to the ability of these two components of oregano essential oil to inhibit inflammation.
References:
1.Capozzi, M.E.; Hammer, S.S.; Mccollum, G.W.; Penn, J.S. Epoxygenated fatty acids inhibit retinal vascular inflammation. Sci Rep 2016, 6, 39211.
2.Yoshihara, T.; Shimada, K.; Fukao, K.; Sai, E.; Sato-Okabayashi, Y.; Matsumori, R.; Shiozawa, T.; Alshahi, H.; Miyazaki, T.; Tada, N.; Daida, H. Omega 3 polyunsaturated fatty acids suppress the development of aortic aneurysms through the inhibition of macrophage-mediated inflammation. Circ. J. 2015, 79, 1470-1478.
5.Bovine immunoglobulins also include IgD and IgE. Why does the manuscript only focus on IgA, IgG, and IgM? Is there any evidence indicating that these three immunoglobulins are more important or have certain traits for castrated bulls?
Response: Thank you very much for your valuable comments. In the mucosal immunity of the digestive tract, IgA forms the first protective barrier of immune defense and is the dominant antibody of mucosal immunity, accounting for more than 15% of the mucosal immunity. IgG accounts for about 80% of the total immunoglobulins in the mucosal immunity and forms the second protective barrier of immune defense [1], and IgM accounts for 5% to 10% of the total immunoglobulins. These three immunoglobulins alone account for more than 98% of the total immunoglobulins. Secondly, IgD content is only 1% of total immunoglobulin, which is easy to polymerize during the isolation process and is highly susceptible to enzymatic cleavage, and the biological function of IgD is not very clear in the current study [2]. IgE accounts for only 0.002% of the total immunoglobulin and is the main antibody involved in the regulation of the pathogenesis of allergic rhinitis, allergic asthma, and eczema, etc. Abbas, B [3] in his study demonstrated that natural infection in cattle infected with Mycobacterium tuberculosis was reflected by the detection of IgA, IgG and IgM. Graham, D A [4] in his study showed that the detection of IgA, IgG and IgM rheumatoid factors in calves had an effect on serology. Costa, A[5] determined that changes in the concentration of IgA, IgG and IgM in colostrum of Holstein cows reflected the immunological level of the Holstein cow. levels. Therefore, in the present study, the immunoglobulin levels of IgA, IgG and IgM could reflect the immunity level of the organism. The physiological functions of IgD and IgE and their roles in oregano essential oil need to be further investigated.
References:
1.Davis, S.K.; Selva, K.J.; Kent, S.J.; Chung, A.W. Serum iga fc effector functions in infectious disease and cancer. Immunol. Cell Biol. 2020.
2.Flemming, A. Uncovering the mystery of secreted igd. Nat. Rev. Immunol. 2018, 18, 668-669.
3.Abbas, B.; Riemann, H.P. Igg, igm and iga in the serum of cattle naturally infected with mycobacterium paratuberculosis. Comp Immunol Microbiol Infect Dis 1988, 11, 171-175.
4.Graham, D.A.; Mawhinney, K.A.; Adair, B.M.; Merza, M. Testing of bovine sera by elisa for igg, igm and iga rheumatoid factors. Vet Immunol Immunopathol 1998, 61, 239-250.
5.Costa, A.; Goi, A.; Penasa, M.; Nardino, G.; Posenato, L.; De Marchi, M. Variation of immunoglobulins g, a, and m and bovine serum albumin concentration in holstein cow colostrum. Animal 2021.
6.In conclusion section, the authors emphasized that oregano essential oil (OEO) regressed the immunoglobulin content towards a normal level, which seem to be inaccurate, because there were no non-castrated bull cells as healthy controls.
Response: Thank you very much for your valuable comments. This question I have modified in the article to say that the addition of oregano essential oil lowered immunoglobulin levels and is highlighted with a yellow background on line 375. I have added the revised conclusion section here for your convenience.
The results of this study showed that the three positive cells were diffusely distributed in the intestinal lamina propria. The addition of OEO to the diets of castrated Holstein bulls did not alter this distribution, but rather enhanced the immunity of these bulls, repaired the damaged immune system, reduced the inflammatory response, and reduced immunoglobulin content. This study provides a theoretical basis for OEO to regulate immunoglobulin content and suppress inflammatory responses.

Reviewer 2 Report
Comments and Suggestions for Authors
This study aimed to investigate the effects of oregano essential oil (OEO) on IgA+, IgG+, 16 and IgM+ cells in the jejunum of castrated Holstein bulls. there are no reports on whether OEO has an immune-boosting effect in castrated Holstein bulls. It adds the effects of OEO in a new group of animals
Authors have to improve the paragraph of material and methods
Line 9: delete Brief summary
Abstract: reduce the abstract, the length must be 200 words
Line 83: add a table showing the composition of oregan extraction oil
Line 88: indicate the statistical test that led to the choice of this sample number
Line 88: better explain how oregano was added to basal diet
Line 89: add a table showing the nutrient (like ADF,NDF,Protein, ash, etc) of the basal diet
Line 90: what does pre-test period means?
Paragraph 2.3: add information (code, manufacter etc) for Hematoxylin and eosin. Add a reference for this technique. Add technical information about the slide scanner. Explain better how define pathological changes
Line 134: add information (code, manufacter etc) for Dapi
Line 168: re stands for respectively?
Figure 4-5-6: add letters into images
Comments on the Quality of English Language
Minor editing of English language required
Author Response
Dear Reviewer:
Thank you for your review and suggestions on our manuscript entitled "Effects of oregano essential oil on IgA+, IgG+ and IgM+ cells in the jejunum of castrated Holstein bulls" (ID: animals-2725639). All these comments are valuable and helpful for us to revise and improve the paper. We have carefully studied these comments and revised them. We hope that they will be recognized. The revised parts are highlighted in the paper with yellow background.
The revised manuscript has been uploaded for your review.
Thank you and with the utmost respect.
Qiyan Liu
The major revisions in the paper and responses to your comments are listed below:
Line 9: delete Brief summary
Response: Thank you very much for your valuable comments, I have removed the above issue and highlighted it with a yellow background.
Abstract: reduce the abstract, the length must be 200 words
Response: Thank you very much for your valuable comments, the abstract has been changed in the article and highlighted with a yellow background. I have added the revised abstract here for your convenience.
Abstract: The aim of this study was to investigate the effect of oregano essential oil on IgA+, IgG+, and IgM+ cells in the jejunum of castrated Holstein bulls. Twelve castrated Holstein bulls were randomly divided into control (YCK) and oregano essential oil (YEO) groups. Pathological changes in the jejunum were observed by HE staining, and the expression levels of IgA, IgG, and IgM in the jejunum were detected by ELISA. The distributions of IgA+, IgG+, and IgM+ cells in the jejunum were analysed by multiplex immunofluorescence and immunohistochemistry. The results showed that the jejunal villi were detached in the YCK group, which may have been related to inflammation, while the intestinal epithelium was clear and intact in the YEO group. The expressions of IgA, IgG, and IgM were significantly reduced by 40.75%, 30.76%, and 50.87%. The IgA+, IgG+, and IgM+ cells were diffusely distributed in the lamina propria of the jejunum, and were reduced by 17.07%, 6.44%, and 6.15%, respectively. Oregano essential oil did not alter the distribution characteristics of IgA+, IgG+, or IgM+ cells in the jejunum, but it suppressed inflammatory response, decreased immunoglobulin content, and significantly enhanced the formation of an immune barrier in the gastrointestinal mucosa.
Line 83: add a table showing the composition of oregan extraction oil
Response: Thank you very much for your valuable comments. The oregano essential oil used in this study was purchased from Ralco Corporation, Marshall, Minnesota, USA, containing 1.3% oregano essential oil and 98.7% natural feed-grade inert carriers. This brand of oregano essential oil has been published by our research group. [1,2]。
References:
- Jia, L.; Wu, J.; Lei, Y.; Kong, F.; Zhang, R.; Sun, J.; Wang, L.; Li, Z.; Shi, J.; Wang, Y.; Wei, Y.; Zhang, K.; Lei, Z. Oregano essential oils mediated intestinal microbiota and metabolites and improved growth performance and intestinal barrier function in sheep. Front Immunol 2022, 13, 908015.
- Zhang, R.; Wu, J.; Lei, Y.; Bai, Y.; Jia, L.; Li, Z.; Liu, T.; Xu, Y.; Sun, J.; Wang, Y.; Zhang, K.; Lei, Z. Oregano essential oils promote rumen digestive ability by modulating epithelial development and microbiota composition in beef cattle. Front Nutr 2021, 8, 722557.
Line 88: indicate the statistical test that led to the choice of this sample number
Response: Thank you very much for your valuable comments. The overall mean and standard deviation were calculated using spss 26.0 to measure the overall weight level and degree of dispersion of the test animals and the result was (350.3±6.49). It indicates that the overall discrete values of the test animals were small. Generally, a standard error of plus or minus 6.49 for large animals is acceptable to many researchers due to the large individual variation in large animals [1]。
References:
- Deng, J.; Zhang, S.; Li, Y.; Shi, C.; Qiu, X.; Cao, B.; He, Y.; Su, H. Effect of Potato Vine and Leaf Mixed Silage Compared to Whole Corn Crops on Growth Performance, Apparent Digestibility, and Serum Biochemical Characteristics of Fattening Angus Bull. Animals (Basel) 2023, 13.
Line 88: better explain how oregano was added to basal diet
Response: Thank you very much for your valuable comments. The basal diet for this experiment was supplemented with oregano essential oil by feeding once a day at 7:00 and once a day at 17:00, weighing 20 g of oregano essential oil each time, mixing it thoroughly with the concentrate and spreading it on the feed surface to ensure complete feed intake by the cows. This part of the article has been modified and highlighted in yellow on line 94.
Line 89: add a table showing the nutrient (like ADF,NDF,Protein, ash, etc) of the basal diet
Response: Thank you very much for your valuable comments. In this experiment, the following tests were done on the nutrient composition of the basal diet, as shown in Table 1, and a table of the basal diet was added in the experimental design section of the main text highlighted with a yellow background on line 105.
Table 1. composition and nutrient levels of the basal feed. [1]
|
Items% |
Months |
||||
|
â… |
â…¡ |
â…¢ |
â…£ |
Ⅴ~Ⅸ |
|
|
Ingredients |
|
|
|
|
|
|
Corn silage |
45.00 |
40.00 |
30.00 |
25.00 |
20.00 |
|
Whole cottonseed |
0.00 |
0.00 |
0.00 |
0.00 |
10.78 |
|
Flattened corn |
14.84 |
10.00 |
8.00 |
5.00 |
10.00 |
|
Corn |
25.17 |
41.36 |
51.06 |
59.93 |
54.36 |
|
Soybean meal |
8.21 |
3.20 |
0.00 |
2.00 |
0.00 |
|
Rapeseed meal |
0.00 |
0.54 |
5.17 |
3.57 |
0.00 |
|
Cottonseed meal |
0.00 |
2.00 |
2.00 |
1.00 |
1.00 |
|
Pea protein powder |
3.49 |
0.00 |
0.00 |
0.00 |
0.00 |
|
Fatty acid calcium |
0.00 |
0.00 |
0.00 |
0.00 |
1.00 |
|
CaHPO4 |
1.12 |
0.80 |
1.10 |
1.28 |
0.65 |
|
Na Cl |
0.62 |
0.30 |
0.47 |
0.50 |
0.47 |
|
NaHCO3 |
1.11 |
1.00 |
1.28 |
1.00 |
1.15 |
|
MgO |
0.00 |
0.19 |
0.28 |
0.16 |
0.13 |
|
Premix1 |
0.44 |
0.61 |
0.64 |
0.56 |
0.46 |
|
Total |
100.00 |
100.00 |
100.00 |
100.00 |
100.00 |
|
Nutrition levels2 |
|
|
|
|
|
|
CP |
11.94 |
10.95 |
10.80 |
10.60 |
9.80 |
|
Total digestible nutrients |
72.10 |
76.54 |
77.51 |
78.74 |
80.52 |
|
Net energy maintenance /(MJ/kg) |
7.44 |
7.97 |
8.14 |
8.31 |
8.53 |
|
Net weight gain / (MJ/kg) |
4.92 |
5.31 |
5.47 |
5.61 |
5.80 |
|
Ca |
0.61 |
0.50 |
0.59 |
0.63 |
0.37 |
|
P |
0.31 |
0.33 |
0.36 |
0.35 |
0.34 |
Note:1 Contained the following per kg of premix: VA 4 000 000 IU, VD3 600 000 IU, VE 120 000 IU, Mn 3 340 mg, Fe 4 800 mg, Zn 12 600 mg, Cu 3 140 mg, Se 60 mg, I 90 mg, Co 22 mg, nicotinic acid 5 970 mg, biotin 200 mg. 2 Total digestible nutrients, Net energy maintenance and Net weight gain were calculated values, while the others were measured values.
References:
- Fan, K.; Li, Z.; Sun, J.; Zhang, X.; Cheng, Q.; Zhang, Z.; Zhang, R.; Jia, L.; Shi, J.; Wang, Y.; et al. Effects of oregano essential oil on growth performance, slaughtering performance and meat quality of Holstein dairy bulls. Chinese Journal of Animal Nutrition 2022, 34, 3012-3022.
Line 90: what does pre-test period means?
Response: Thank you very much for your valuable comments. In this study, the pre-experiment is the acclimatization phase, which is designed to prevent stress in the test animals during the formal test to ensure the accuracy and stability of the test. The word "pre-test" has been changed to "acclimatization" in the text and highlighted in line 88 with a yellow background.
Paragraph 2.3: add information (code, manufacter etc) for Hematoxylin and eosin. Add a reference for this technique. Add technical information about the slide scanner. Explain better how define pathological changes
Response: Thank you very much for your valuable comments. Hematoxylin was purchased from Beijing Solepol Co., Ltd, No. H8070; Eosin was purchased from Beijing Solepol Co., Ltd, No. G1102; Slide Scanner was purchased from Shandong Sretian Medical Technology Co., Ltd, Model: DX1. Pathologic changes are invisible changes in specific areas due to the action of pathogenic factors and the body's pathological response. The site, nature and extent of the disease can be understood by observing these changes. In this study, we found that almost all jejunal villi in YCK group were severely damaged and almost all of them were detached, and the local central celiac duct was filled with pink plasma material, the central celiac duct was dilated, and more neutrophils were seen around the intestinal line, and the intestinal villi were obviously edematous in the lamina propria. This part has been modified and highlighted in paragraph 3.1 with a yellow background.
Line 134: add information (code, manufacter etc) for Dapi
Response: Thank you very much for your valuable comments. DAPI was purchased from Beijing Solepol Technology Co., Ltd. under item number C0065 and is highlighted on line 144 with a yellow back ground. Line 168: re stands for respectively?
第 168 行:re stands for respectively?
Response: Thank you very much for your valuable comments. The reduction rate and the table's reference to the Decrease rate, which represents the rate of reduction, have been changed in the text from "reduction rate" to "Decrease rate" and highlighted in line 190 with a yellow background.
Figure 4-5-6: add letters into images
Response: Thank you very much for your valuable comments. Figure 4-5-6 has been re-uploaded.
